# Circulating and Intracellular miRNAs as Prognostic and Predictive Factors in HER2-Positive Early Breast Cancer Treated with Neoadjuvant Chemotherapy: A Review of the Literature

**DOI:** 10.3390/cancers13194894

**Published:** 2021-09-29

**Authors:** Chrystel Isca, Federico Piacentini, Ilenia Mastrolia, Valentina Masciale, Federica Caggia, Angela Toss, Claudia Piombino, Luca Moscetti, Monica Barbolini, Michela Maur, Massimo Dominici, Claudia Omarini

**Affiliations:** 1Division of Medical Oncology, Department of Medical and Surgical Sciences for Children & Adults, University Hospital of Modena, 41124 Modena, Italy; 231204@studenti.unimore.it (C.I.); federico.piacentini@unimore.it (F.P.); federica.caggia@unimore.it (F.C.); angela.toss@unimore.it (A.T.); 256171@studenti.unimore.it (C.P.); mbarboli@unimore.it (M.B.); massimo.dominici@unimore.it (M.D.); 2Laboratory of Cellular Therapy, Department of Medical and Surgical Sciences, University of Modena and Reggio Emilia, 41124 Modena, Italy; ilenia.mastrolia@unimore.it (I.M.); valentina.masciale@unimore.it (V.M.); 3Division of Medical Oncology, Department of Oncology-Hematology, University Hospital of Modena, 41124 Modena, Italy; Moscetti.luca@aou.mo.it (L.M.); maur.michela@aou.mo.it (M.M.)

**Keywords:** exosome, miRNA, extracellular vesicles, HER2 positive breast cancer, early stage breast cancer, neoadjuvant chemotherapy, trastuzumab, predictive factors, prognostic factors

## Abstract

**Simple Summary:**

Breast cancer is a leading cause of female cancer-related death worldwide. Anti-HER2-targeted therapies dramatically improved prognosis for HER2-positive breast cancer patients. Despite that, growing drug resistance due to the pressure of therapy is relatively frequent. For that reason, it is necessary to find biomarkers able to predict treatment sensitivity and survival outcomes. Increasing research has shown how miRNAs, secreted by tumor cells, are strongly involved in cancer development. In this review, we will discuss the recent evidence on the predictive and prognostic value of miRNAs involved in HER2-positive early breast cancer progression.

**Abstract:**

MicroRNAs (miRNA) are small noncoding RNAs that can act as both oncogene and tumor suppressors. Deregulated miRNA expression has been detected in human cancers, including breast cancer (BC). Considering their important roles in tumorigenesis, miRNAs have been investigated as potential prognostic and diagnostic biomarkers. Neoadjuvant setting is an optimal model to investigate in vivo the mechanism of treatment resistance. In the management of human epidermal growth factor receptor-2 (HER2)-positive early BC, the anti-HER2-targeted therapies have drastically changed the survival outcomes. Despite this, growing drug resistance due to the pressure of therapy is relatively frequent. In the present review, we focused on the main miRNAs involved in HER2-positive BC tumorigenesis and discussed the recent evidence on their predictive and prognostic value.

## 1. Introduction

Breast cancer (BC) is the most common type of cancer diagnosed in women worldwide, with around 2.3 million new cases in 2020 and 685,000 deaths globally according to the World Health Organization [1]. The human epidermal growth factor receptor-2 (HER2) is overexpressed in almost 20% of cases and is generally associated with poorer survival outcomes [2]. Thanks to the advent of anti-HER2-targeted therapies, prognosis for HER2-positive BC patients has dramatically improved [3,4,5,6,7,8,9,10,11,12]. Despite their objective activity, intrinsic and acquired resistance to anti-HER2 agents remains a major clinical challenge [13]. To address this problem of resistance, considerable efforts have been made to investigate the mechanism of treatment failure. The neoadjuvant setting is an optimal model for studying in vivo the mechanism of treatment resistance [14,15].

Growing evidence has indicated that microRNA (miRNA) profiling could be a promising biomarker for early diagnosis, treatment sensitivity/resistance, and prognosis in different cancer types, including BC [16,17,18]. MiRNAs are a class of small noncoding regulatory RNAs that are involved in both physiological and pathological processes, such as cancer proliferation [19]. miRNAs can be easily detected in blood and tumor tissue [19]. In particular, miRNAs secreted by cancer cells are packed into extracellular vesicles (EVs) and released into peripheral blood. EVs are largely involved in the cross-talk between cancer cells and the microenvironment [20,21]. The important involvement of EVs and miRNA in ontogenetic processes and their availability in biological fluids, make these molecules optimal biomarkers in oncology [20,22].

This review summarizes the main miRNAs involved in HER2-positive BC tumorigenesis, focusing on their prognostic and predictive role.

## 2. Extracellular Vesicles and MicroRNAs

EVs are small lipid bilayer-delimited particles that are naturally released by cells and involved in cell communication [23]. EVs can be detected in biological fluids such as serum, plasma, urine, saliva, breast milk, bile, amniotic fluid, and cerebrospinal fluid [24,25]. Once released into the extracellular fluid, EVs may link to the cell plasma membrane, fusing directly with the membrane or being endocytosed, resulting in delivery of proteins and RNA into the target cell [24]. EVs are involved in several biological processes such as antigen presentation, immune regulation, apoptosis evasion, drug resistance, and angiogenesis [25,26,27,28]. Based on their size, they are classified into apoptotic bodies (50–5000 nm) released by apoptotic cells, shedding microvesicles/ectosomes (100–1000 nm) produced by budding from the plasma membrane, and exosomes (30–150 nm) that originate from intracellular multivesicular bodies and are secreted in the extracellular space upon fusion with the plasma membrane [20,23,29]. They carry proteins, metabolites, lipids, mRNA, miRNA, long-non-coding RNA, and DNA [24,25,26,30,31]. A large body of evidence suggests that cancer cells release a higher amount of EVs compared to non-malignant cells [25,26]. The most common methods reported for isolating exosomes from serum or plasma are commercial kits available for the isolation of EVs the size of exosomes (e.g., Total Exosomes Isolation Kit—Invitrogen) [29]. Exosomes contain miRNAs that are strongly involved in the cross-talk between cancer cells and the microenvironment. miRNAs are short, noncoding, single-stranded RNAs of 21–25 nucleotides, which regulate gene expression at a post transcriptional level by binding to the 3′untranslated region (3′UTR) of its target messenger RNA [32]. miRNAs can be present in the peripheral blood or packed into EVs such as exosomes [29]. MiRNAs are a class of small, noncoding, regulatory RNAs that are involved in both physiological and pathological processes, such as cancer proliferation. In physiological processes, miRNA are produced by non-cancerous cells. The most common techniques reported for miRNA profiling are: microarray analysis; bead-based flow-cytometric techniques; and real-time PCR, SAGE, or RAKE assay, with peculiar strengths and weaknesses under investigation [33,34,35,36,37].

## 3. miRNAs in HER2-Positive Breast Cancer

Several preclinical and clinical studies have focused their attention on the research of miRNAs linked to HER2 pathways in BC. In most of these cases, miRNAs were isolated from tumor tissues (cell lines or formalin-fixed paraffin embedded samples). Table 1 summarizes all the miRNAs known to be involved in HER2-positive BC tumorigenesis. The most widely reported miRNAs in the literature are: miR-155, miR-21, miR-205, and miR-125 families. In the literature, the most widely cited miRNAs related to HER2-positive subtypes are the following:

miR-155 has been found upregulated in HER2-positive BC in both early and metastatic settings [38]. miR-155 has an oncogene function in the interaction with the MAPK pathway through constitutive activation of STATs proteins by inhibiting *SOCS1* gene and activating AKT and Src [39,40].

miR-21 is one of the most investigated miRNAs in oncology. It has been found overexpressed in all BC subtypes, but mainly in the HER2-positive hormone receptors negative subtype [41,42]. In particular, high expression of miR-21 has been found to significantly correlate with more aggressive tumor behavior (larger tumor with high nuclear grade) and poor survival outcomes [43]. In vitro studies have shown that up-regulation of miR-21 induced by HER2/neu signaling via MAPK (ERK1/2) pathway promotes cell invasion through downregulation of PDCD4 (programmed cell death 4) [43]. 

miR-205 has been detected in serum of BC patients with a decreasing expression from the less aggressive BC subtypes to the more aggressive ones, mainly in HER2-positive and triple-negative tumors [44,45]. It targets HER3 directly, inhibiting SKBr3 in BC cell lines [46,47]. Its oncosuppressive role is also linked by EMT suppression [45]. 

Its downregulation in serum of BC patients compared to healthy people suggests a possible role as a marker for BC diagnosis [63,64].

miR-125b belongs to the miR-125 family. miR-125b is generated from two genes: miR-125b-1 (on chromosome 11q24) and miR-125b-2 (on chromosome 21q21). It can be up- or downregulated depending on the tumor type [69,70,71,72]. In BC patients, its upregulation seems to promote EMT expression with an important implication in metastasis development [73]. Other direct targets of miR-125b are the EGFR family genes, mainly ERBB2 and ERBB3. In particular, in BC cell lines, miR-125b seems to induce a suppression of these genes [74]. Evidence from paraffin-embedded BC tissue samples reported a significant correlation between miR125b expression and HER2 expression (*p* < 0.001), with an important prognostic implication for the patient’s overall survival (OS) [66]. 

Finally, Lowery et al, through the expression profiling analysis of 453 miRNAs from formalin-fixed paraffin-embedded samples of 29 early BC patients, using an artificial neural network (ANN) and RQ-PCR for validation, reported a signature of five miRNAs (miR-520d, miR-181c, miR-302c, miR-376b, and miR-30e-3p), which seemed to accurately predict HER2 status in early BC patients [51].

## 4. Predictive Biomarkers

The identification of biomarkers able to predict treatment sensitivity before any intervention remains one of the main goal for oncologists. Thanks to the easy availability of miRNAs, they can be an optimal biomarker. Several published papers investigated the predictive role of miRNAs isolated in both BC tissue and blood. Table 2 summarizes miRNAs known to be predictive of chemotherapy or anti-HER2-targeted therapy sensitivity. In particular, data from different clinical and preclinical studies identified miR-125b, miR21, miR-210, and miR-155 as predictive biomarkers of trastuzumab resistance. Circulating miR-125b in blood serum of early BC patients seems to be significantly associated with neoadjuvant chemotherapy response. In particular, higher expression levels of miR-125b have been reported in resistant patients compared to responsive ones [75]. Wang et al. demonstrated an association between higher miR-125b expression and a higher percentage of proliferating cells in BC tissue [75]. Biologically, it can be explained by the Bak1 (Bcl-2 antagonist Killer 1) downregulation induced by miR-125b, which suppresses drug efficacy [76]. BaK1 is also known to be involved in the epithelial–mesenchymal transition process through EMT dysregulation, resulting in an increased resistance to chemotherapy [77]. In addition, preclinical evidence has suggested that the upregulation of miR-21 induced trastuzumab resistance due to the reduction of *PTEN* gene expression [78,79,80]. Another possible biological mechanism of resistance is PI3K pathway dysregulation by miR-21 and EMT activation with the triggering of IL-6/STAT3/NF-kB-mediated signaling [81]. A study by Liu et al on circulating miR-21 (ser-mir21) in HER2-positive early BC patients treated with neoadjuvant chemotherapy plus trastuzumab showed a significant correlation between changes in ser-miR-21 expression and treatment sensitivity [78]. The authors reported a significant reduction of expression of ser-miR-21 levels at the end of the second cycle and at the end of neoadjuvant treatment compared to the baseline value (before start of treatment), mainly shown in clinical responders (*p* < 0.001). These results were confirmed by Rodríguez-Martínez et al., who observed lower levels of miR-21 in HER2-positive patients during neoadjuvant treatment with trastuzumab [79]. They also highlighted a significant association between higher levels of exosomal miR-21 and circulating tumor cells [79]. In a study by Jung et al. on plasma samples of BC patients who underwent neoadjuvant trastuzumab-based chemotherapy, levels of miR-210 inversely correlated with treatment response. Higher levels of miR-201 have been reported in patients with residual disease compared to those with pathologically complete response (*p* = 0.0359) [82]. An in vitro study confirmed these data, showing a higher expression of miR-21 in trastuzumab-resistant BT474 cells [82]. miR-155 was identified as a negative predictive marker too. In a cohort of 175 BC patients, 107 early stage and 68 in the metastatic setting, the upregulation of miR-155 was related to poor trastuzumab sensitivity [83]. 

On the contrary, miR-148a-3p and miR-205 seem to be positive predictive biomarkers for anti-HER2-targeted therapies. In particular, the early blood detection of miR-148a-3p seems to identify early responsive patients to neoadjuvant anti-HER2 treatments [100]. Indeed, the study by Di Cosimo et al. demonstrated a strong correlation between miR-148a-3p and a pathologically complete response in a BC population treated with primary trastuzumab (*p* = 0.008) [100]. Consistently, in vitro studies reported that miR-148a over-expression inhibits BC cell proliferation through inhibition of the MAPK/ERK signaling pathways by direct targeting of ERBB3 genes and angiogenesis inhibition [98]. Preclinical evidence has shown how the miR-205 restores a pro-apoptotic activity. In BC tissues, miR-205 directly targets HER3 receptor and inhibits the *AKT*-mediated survival pathway, increasing the responsiveness to both tyrosine kinase inhibitors (such as Lapatinb and Gefitinb) and trastuzumab [47,101]. Finally, a recent preclinical study showed a significant dysregulation of four miRNAs (miR-23b-3p, miR-195-5p, miR-656-5p, miR-340-5p) in HER2-positive BC-resistant BT-474 cells, suggesting potential involvement of these miRNAs in trastuzumab resistance mechanisms [91]. 

## 5. Prognostic Biomarkers

As well as the predictive role of miRNAs being widely investigated, their prognostic value has also been studied (Table 3). The most widely reported miRNAs as negative prognostic biomarkers are: miR-21, miR-155, miR-150-5p, and miR-4734. miR-21 demonstrated an important prognostic role in HER2-positive early BC patients receiving neoadjuvant chemotherapy combined with trastuzumab [78]. In particular, Liu et al. demonstrated a better OS and disease-free survival (DFS) in patients with decreased ser-miR-21 from the start to the end of neoadjuvant treatment [78]. Higher levels seemed to be associated with larger tumor size and high Ki67 expression, with a statistically significant correlation with higher stage (*p* = 0.008) and tumor grade (*p* = 0.005). In particular, HER2-positive BC subtype (*p* = 0.002) and negative estrogen receptor cancer (*p* = 0.002) had a significantly higher expression of miR-21 [41]. Moreover, in a study on serum samples from 127 patients with early HER2-positive BC undergoing neoadjuvant treatment, increased levels of circulating miR-21 before and after chemotherapy showed a significant association with poor OS [84]. In early BC, higher miR-155 expression showed a strong correlation with poor event-free survival too, in univariate and multivariate analyses [83]. Du et al. identified two intracellular miRNAs, miR-150-5p and miR-4734, as reliable prognostic biomarkers predicting development of recurrence disease in HER2-positive BC after adjuvant trastuzumab-based treatment [102]. 

Contrasting clinical roles have been reported for miR-125. In particular, miR-125-b was identified as a negative prognostic factor, associated with worse OS in HER2-positive BC patients [66]. On the contrary, miR-125a-5p was a positive prognostic factor. A low expression level was found in the serum of patients with shorter survival compared to long-term patients. It also had a negative correlation with tumor grade (*p* = 0.004), lymph-node status (*p* = 0.004), and tumor size (*p* < 0.001). [103]. 

Both miR-205 and miR-148a seem to be positive prognostic factors. In a clinical study on 52 HER2-positive BC patients treated with adjuvant trastuzumab, miR-205 expression on sample tissues significantly correlated to better DFS (*p* = 0.00168) [101]. In addition, in cell lines, lower levels of miR-205 were correlated to high histological grade biopsies and higher invasion rate [104]. In preclinical evidence, miR-148a inhibits BC cell migration and invasion, through wnt-βcatenin pathway inhibition [105] and MMP-13 downregulation [106], resulting in a pro-apoptotic activity [107]. Among positive prognostic factors, miR-145 has been reported by Quan et al. as downregulated in tumor samples more than in paracancerous tissue [108]. In agreement with its oncosuppressive function, higher miR-145 levels seem to be associated with a better survival rate than the lower expression group (*p* < 0.028) [48,108]. 

As expected, due to the close connection between predictive and prognostic value, some miRNAs overlap between Table 1 and Table 2, specifically, miR-210, miR-205, miR-21, miR-125b, and miR-148. High expression levels of miR-210, miR-21, and miR-125b correlate to trastuzumab and/or chemotherapy resistance and to poor survival, accordingly [66,75,78,82,84]. On the contrary, high expression of miR-205 and miR-148 correlated to an increase in sensitivity to cytotoxic treatments and a higher number of pathologically complete responses at the end of the neoadjuvant therapy, with an improvement in OS [104,105,106,107].

## 6. Conclusions

Emerging evidence highlights the key role in cell-to-cell communication and tumorigenesis of miRNAs. Due to their easy availability in blood and tumor tissue, they represent the optimal biomarker in oncology. Actually, a great deal of effort has been put into basic and clinical research to isolate a signature of miRNAs with predictive and prognostic value. This issue is particularly important in patients treated with targeted therapies, such as HER2-positive BC therapies. Data from our review identified a panel of miRNAs expressed in HER2-positive BC potentially able to predict treatment and survival outcomes. Unfortunately, available data are not enough to define whether some miRNAs are better predictors than others. In the available studies, the miRNA level changes were evaluated in different ways, using fold change in some cases and p-value in others, which made a comparison between the studies difficult. Moreover, further investigations would be necessary to properly define and validate these molecular tools as clinical biomarkers. Despite this, our literature review identified a triplet of miRNAs (155, 125b, 21), typically expressed in HER2-positive BC, that seem to be valid negative predictive and prognostic biomarkers. Available evidence is promising but requires further data from clinical trials. A future development could be the periodic assessment of a panel of HER2-specific miRNAs in patients on neoadjuvant treatment to enable early prediction of treatment resistance and to tailor patient management.

## Figures and Tables

**Table 1 cancers-13-04894-t001:** HER2-positive breast cancer miRNA expression.

miRNA	Expression Level	Gene Target ^#^	Action	Evidence	Sample	References
miR-489	down	SMAD3, SHP2, HER2	Cell growth, invasion and EMT inhibition (decreased expression of HER2)	clinical/preclinical	tissue	[48]
miR-520d	down	TWIST1	Restoration of E-cadherine expression (reduction in invasiveness)	clinical/preclinical	tissue	[49,50]
miR-376b	up	Hoxd10	Angiogenesis	clinical/preclinical	tissue	[51,52]
miR-342-5p	down	ERK, MAPK, SAPK/JNK	HER2 downstream inhibiting effect	preclinical	tissue	[53]
miR-375	up/down	PAX6, IGF1R	Tumor suppressor (HER2, HR *-negative and inflammatory BC ^§^-related)	clinical/preclinical	tissue/blood	[54,55,56]
miR-155	up	SOCS1, JAK2/STAT3, FOXO3	HER2-positive status and TNBC correlation (cell proliferation)	clinical/preclinical	tissue	[38,39]
miR-181c	down	OPN	Increased HER2-positive signature, enhanced chemosensitivity	clinical/preclinical	tissue	[51,57,58]
miR-302c	up	MEKK1	Increased HER2-positive signature, enhanced chemosensitivity	clinical/preclinical	tissue	[51,59]
miR-30e-3p	up/down	CTHRC1	Increased HER2-positive signature, proliferation inhibition, and apoptosis	clinical/preclinical	tissue	[51,60]
miR-200	up/down	EGFR, SIRT1	HER2-positive and HR * negative status association	clinical/preclinical	tissue	[61,62]
miR-205	down	EGFR, VEGF-A, HER3, ZEB1, SIP1, E2F	Tumor suppressor (cell cycle arrest and apoptosis)	clinical/preclinical	tissue/blood	[45,63,64]
miR-122-5p	up	ADAM10	Tumor suppressor (cell cycle arrest and apoptosis)	preclinical/clinical	tissue	[65]
miR-125b	up	ERBB2,BAK1, BCL-2, STARD13	Increased migration and invasion	clinical	tissue	[66]
miR-125a-5p	down	EGFR family, AKT-ERK	Tumor suppressor (invasion reduction)	clinical/preclinical	tissue	[61]
miR-21	up	PDCD4, PTEN, TPM1, MAPK, EGFR family	HER2/neu upregulation correlation and apoptosis inhibition	clinical/preclinical	tissue	[41,61]
miR-147	down	EGFR1, AKT2, CDK4, RB1	EGFR-driven cell-cycle (cell proliferation inhibition)	preclinical	tissue	[67]
miR-124	down	EGFR1, STAT3	Cell growth and differentiation inhibition	preclinical	tissue	[68]

^#^ Gene targets included in the table were identified through an in silico analysis; * HR: hormone receptors; ^§^ BC: breast cancer.

**Table 2 cancers-13-04894-t002:** Predictive value for treatment outcomes of miRNAs expressed in HER2-positive early breast cancer.

miRNA	Expression Level	Gene Target ^#^	Action	Evidence	Sample	References
miR-210	up	RAD52, FGFR1, E2F3, ephrin A3, MET, IGFR1, MUC4	Trastuzumab resistance	preclinical/clinical	tissue/blood	[82,84]
miR-375	down	IGF1R, JAK2/STAT3	Trastuzumab, tamoxifene, and adriamycin resistance	preclinical/clinical	tissue/blood	[55,56,85]
miR-205	up	EGFR, VEGF-A, HER3, AKT	Increased sensitivity to anti-HER2 therapy	preclinical	tissue	[61]
miR-125	up	ERBB2, BAK1, BCL-2, STARD13	Increased sensitivity to anti-HER2 therapy	preclinical	tissue	[61]
miR-21	up	NFKB, MAPK, PTEN	Trastuzumab and taxol resistance	preclinical/clinical	tissue/blood	[78,79,80,81]
miR-125b	up	ERBB2, BAK1, BCL-2	Resistance to anthracycline/taxol-based therapies	preclinical/clinical	tissue/blood	[75]
miR-30c	up	TWF1, IL-11, YWHAZ	Increased sensitivity to taxol, doxorubicin and tamoxifene	preclinical	tissue	[86,87]
miR-26a	up	CCNE2	Increased sensitivity to trastuzumab	preclinical	tissue	[88]
miR-30b	up	CCNE2	Increased sensitivity to trastuzumab	preclinical	tissue	[88,89]
miR-34a	down	BCL-2, CCND1, NOTCH1	RT sensitivity, multidrug resistance	preclinical/clinical	tissue	[90]
miR-122-5p	up	ADAM10	Increased sensitivity to trastuzumab	clinical	tissue	[65]
miR-23b-3p	up	PTEN	Trastuzumab resistance	preclinical	tissue	[91]
miR-195-5p	down	VEGFA, RAF 1, FGF2, AKT3, CCND1, MYB	Trastuzumab resistance	preclinical	tissue	[91]
miR-656-5p	down	CREB1	Trastuzumab resistance	preclinical	tissue	[91]
miR-340-5p	down	C-MET	Trastuzumab resistance	preclinical	tissue	[91]
miR-770-5p	up	PI3K, MAPK	Potentiated trastuzumab effect	preclinical	tissue	[92]
miR-200b	up	PTEN	Trastuzumab resistance	preclinical/clinical	tissue/blood	[93]
miR-135b	up	CyclinD2	Increased sensitivity to trastuzumab	preclinical/clinical	tissue/blood	[93,94]
miR-29a	up	IGF-1R, PTEN	Trastuzumab and adryamicin resistance	preclinical/clinical	tissue/blood	[95,96]
miR-141	down	ERBB4	Trastuzumab resistance	preclinical	tissue	[97]
miR-148	up	MAPK/ERK (ERBB3 genes)	Increased pathological complete response	clinical	blood	[98]
miR-155	up	SOCS1,STAT3	Trastuzumab resistance	preclinical/clinical	tissue/blood	[39,83]
miR-221	up	PTEN	Trastuzumab resistance	preclinical	tissue	[99]

^#^ Gene targets included in the table were identified through an in silico analysis.

**Table 3 cancers-13-04894-t003:** Prognostic value of miRNAs expressed in HER2-positive early breast cancer.

miRNA	Expression Level	Pos/Neg	Action	Evidence	Sample	References
miR-21	up	neg	Reduction during treatment: better OS and DFS (high level in high histological grade and invasion rate)	preclinical/clinical	tissue/blood	[41,78,84]
miR-122	up	neg	Circulating higher levels associated with metastasis	clinical	tissue/blood	[54,109]
miR-155	up	neg	High levels predicts poor event free/progression free survival	clinical	blood	[83]
miR-205	up	pos	Low level associated with worse prognosis (high level in high histological grade and invasion rate)	preclinical	tissue	[104]
miR-30a	up	pos	Low level associated with higher grade and TNM stage	clinical	tissue	[110]
miR-210	UP	neg	Reduced RFS	preclinical	blood	[84]
miR-125b	up	neg	Worse OS	clinical	tissue	[66]
miR-125a-5p	up	pos	Low levels associated with poor survival, high grade, lymph node status, and tumor size	preclinical/clinical	tissue/blood	[103]
miR-10b	up	neg	Correlation with tumor stage and lymph node positivity	preclinical/clinical	tissue	[111,112]
miR-373	up	neg	Correlation with advanced clinical stage	preclinical	tissue/blood	[84,113]
miR-148a	up	pos	Migration and invasion cancer cell inhibition (pro-apoptotic function)	preclinical	tissue	[105,106,107]
miR-145	up	pos	Low levels correlate with poor survival outcomes	preclinical/clinical	tissue	[108,114,115]
miR-4734	up	neg	Disease recurrence (ERBB2 amplification)	preclinical/clinical	tissue	[102,116]
miR-150-5p	up	neg	Disease recurrence (enhanced cell proliferation, invasion, and migration)	preclinical/clinical	tissue	[102,117]
miR-489	down	neg	Low level correlates with shorter OS	clinical	blood	[118]
miR-770-5p	up	pos	Cell invasion and motility inhibition	preclinical	tissue	[92]
miR-342-5p	up	pos	Better OS and increased time to progression (cell proliferation inhibition)	preclinical/clinical	tissue	[53,119]

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
