# Peer review of "Circulating and Intracellular miRNAs as Prognostic and Predictive Factors in HER2-Positive Early Breast Cancer Treated with Neoadjuvant Chemotherapy: A Review of the Literature"

_cancers, 2021, doi:10.3390/cancers13194894_

Round 1

Reviewer 1 Report

The authors perform a literature review of the studies regarding the use of miRNAs for prediction or prognosis of HER2+ breast cancer. Their main focus is on studies performed in tumor tissue or blood.

Lines 30-32: The authors state that the expression of HER2 is associated to poorer survival outcomes. This statement needs clarification. They should state whether the poorer outcomes are observed when compared to breast cancer overall. It is known that among the principal molecular breast cancer subtypes, triple-negative has the shown the worst outcomes.

Line 36: It is not clear what the authors mean by how cancer cells lead to treatment failure.

Line 36: Please clarify the term neoadjuvant setting.

Line 43: miRNAs are not exclusively secreted by cancer cells. The authors should acknowledge that miRNAs are produced by non-cancerous cells too.

Line 66: The Total Exosome Isolation kit is not the only one suited for exosome isolation. The authors should consider using the term commercially available exosome isolation kits and maybe include the name of the kit as a example.

Line 81: The authors state that they present the most reported miRNAs on Table 1. What was the criteria used for this selection? If it was the number of studies reporting the same targets, there are other candidates reported in two or more studies. Please clarify.

Table  1: It would be useful for the authors to include the type of cohort or each study, whether it was case-control or which groups were used to establish comparisons  (i.e. among different subtypes). This will also allow the reader to know in comparison to which group the expression levels are up or down. It is not clear whether the targets included on the table were validated on each study or were identified through an in silico analysis.

Table 2: Check the title of the table for accuracy regarding the content.  Maybe include “Predictive value for treatment outcomes…”

Section 5: The authors should state the criteria for the special focus on those four miRNAS from the group included in Table 3.

Table 3: The authors could include the study groups used for each study to provide a better understanding of the findings.

Conclusions: The authors could include a diagram of miRNAs that overlap among the endpoints covered on the review (i.e miR-21, miR-155). Also, they could provide a potential explanation for this overlap.

Reviewer 2 Report

The review manuscript by Chrystel et al describes the current knowledge of miRNAs known to be potential biomarkers of HER2 positive breast cancer treated with neoadjuvant chemotherapy. This review systematizes prognostic versus predictive markers of HER2 positive early breast cancers. 

First, it is not clear what the difference between prognostic and predictive biomarkers is as presented. As shown in the two tables listing these biomarkers, some miRNAs appear in both lists. Maybe authors should clarify this. One way to clarify this is to show differences, if any, between miRNAs that are both prognostic and predictive versus only prognostic or only predictive. A null hypothesis (that is, a potential concern) is that these two distinctions are not real and simply come from studies that were looking for specific designations.

Related to the above point, rather than listing miRNA targets in tables, are there any commonalities among targets for each category of miRNAs? In other words, some functional interpretation is needed here.

Second, it is unclear what is meant by “predictive value” such as in the table titles.

Third, more information is needed as to when a miRNA is deemed predictive or prognostic, that is, how an increase or decrease of its levels was assessed. Was is a 10% increase or 100% increase and compared to what marker? These tables should probably have this information.

The outcome here would be to at least attempt to classify these potential biomarkers based on their relative predictive values such as fold change and/or p-value. A list is good, but the key is whether any miRNAs are better predictors than others.

As part of the above point, more details could be used in descriptions such as “ significant correlation between changes in ser-mir-21 expression and treatment sensitivity”. By itself, this phrase is meaningless, but can be meaningful when compared to other studies.

Is there any difference in the cohorts of inter versus intracellular miRNAs? Not sure if this distinction is even important given that some studies are cited as having been done on FFPE samples.

A section is dedicated to extracellular vesicles. Is there a known meaningful difference between different EV particles such as exosomes, microvesicles, apoptotic bodies, in terms of miRNA profiles?

There are many typos and/or English inaccuracies throughout the text.

It is unclear how this study relates to some uncited reviews such as PMCID: PMC7554858

Reviewer 3 Report

This is a review on the role of miRNA in HER2+ early breast cancer. Obviously the topic is much studied. There are no direct clinical applications to date, although I agree with the authors that pathophysiology insights that ware generated in these studies may potentially be important. 

I recommend altering the Conclusions: "Available data suggest that a periodic assessment of a panel of HER2-specific miRNAs in patients on neoadjuvant treatment may be useful to early predict treatment resistance and tailor patient management based on tumor biology". This statement is not supported by any data and there are no trials tailoring treatment based on miRNA.  They can only suggest such a trial and maybe propose the panel that should be done and in what intervals.

Round 2

Reviewer 1 Report

The authors have addressed most of the comments made on the previous round of review. I believe that this have strengthened their work, mostly the conclusions of the literature review.  However, some comments from the previous review are still pending:

Line 41: The sentence was not rewritten as the authors state on the response to reviewers. Please update the modification on the final version of the manuscript as presented on the response.

Table  1: The authors did not provide information regarding the gene targets included in the table, whether these were validated on each study or were identified through an in silico analysis.

Reviewer 2 Report

There have been edits made that lead to improvement. English is still not very good and the manuscript requires extensive editing, especially in introduction/abstract and concluding sections. One example out of many: what is "upcoming drug resistance"? This makes no sense in English. The middle part is described better. 

Overall the conclusions are very confusing. One, there is still no clear distinction made or described between prognostic and predictive biomarkers. It seems like a biomarker is classified in one versus other category merely based on whether a given study included patients treated with drugs or not. The tables are okay in their revised forms, but there should be a better explanation, "commentary", about what these tables mean. Can a biomarker be both predictive and prognostic? By definition, if I understand this correctly, it is not possible.

Authors need to comment on when a biomarker is potentially useful and what miRNAs have been used or are closest to being used as biomarkers for individual patients. The problem is that statistical differences observed between cohorts of patients described here is not sufficient to call it a biomarker because there is much less power to ascertain outcomes for individual patients. Statistical criteria to use miRNA levels for prognosis of individual patients' outcomes are much stricter than statistical differences in cohort analyses. The latter can not be used as basis for biomarkers because they have little value for individual patients. In other words, an issue of statistical differences in levels between groups of patients and prognostic value for individual patients should be clarified.

Round 3

Reviewer 2 Report

The language is up to par now. Authors give a very clear definition of prognostic versus predictive in response to my questions. Those characteristics are very clear and are, by definition, mutually exclusive. Their groupings in the text are anything but. In any case, this should  should be up to the reader to judge any further.